# Bio-Inspired Fiber Reinforcement for Aortic Valves: Scaffold Production Process and Characterization

**DOI:** 10.3390/bioengineering10091064

**Published:** 2023-09-09

**Authors:** Christian A. Boehm, Christine Donay, Andreas Lubig, Stephan Ruetten, Mahmoud Sesa, Alicia Fernández-Colino, Stefanie Reese, Stefan Jockenhoevel

**Affiliations:** 1Department of Biohybrid & Medical Textiles (BioTex), AME Institute of Applied Medical Engineering, Helmholtz Institute, RWTH Aachen University, Forckenbeckstr. 55, 52074 Aachen, Germany; boehm@ame.rwth-aachen.de (C.A.B.); christine.neusser@rwth-aachen.de (C.D.); andreas.lubig@rwth-aachen.de (A.L.); fernandez@ame.rwth-aachen.de (A.F.-C.); 2Electron Microscopy Facility, University Hospital Aachen, Pauwelstr. 30, 52074 Aachen, Germany; sruetten@ukaachen.de; 3Institute of Applied Mechanics, RWTH Aachen University, Mies-van-der-Rohe-Str. 1, 52074 Aachen, Germany; mahmoud.sesa@ifam.rwth-aachen.de (M.S.); stefanie.reese@ifam.rwth-aachen.de (S.R.); 4Aachen-Maastricht Institute for Biobased Materials, Maastricht University at Chemelot Campus, Urmonderbaan 22, 6167 Geleen, The Netherlands

**Keywords:** non-woven scaffold, biohybrid heart valve, fiber reinforcement, e-spinning

## Abstract

The application of tissue-engineered heart valves in the high-pressure circulatory system is still challenging. One possible solution is the development of biohybrid scaffolds with textile reinforcement to achieve improved mechanical properties. In this article, we present a manufacturing process of bio-inspired fiber reinforcement for an aortic valve scaffold. The reinforcement structure consists of polyvinylidene difluoride monofilament fibers that are biomimetically arranged by a novel winding process. The fibers were embedded and fixated into electrospun polycarbonate urethane on a cylindrical collector. The scaffold was characterized by biaxial tensile strength, bending stiffness, burst pressure and hemodynamically in a mock circulation system. The produced fiber-reinforced scaffold showed adequate acute mechanical and hemodynamic properties. The transvalvular pressure gradient was 3.02 ± 0.26 mmHg with an effective orifice area of 2.12 ± 0.22 cm^2^. The valves sustained aortic conditions, fulfilling the ISO-5840 standards. The fiber-reinforced scaffold failed in a circumferential direction at a stress of 461.64 ± 58.87 N/m and a strain of 49.43 ± 7.53%. These values were above the levels of tested native heart valve tissue. Overall, we demonstrated a novel manufacturing approach to develop a fiber-reinforced biomimetic scaffold for aortic heart valve tissue engineering. The characterization showed that this approach is promising for an in situ valve replacement.

## 1. Introduction

Cardiovascular diseases are widespread in the Western world and rank among the most frequent causes of death. Apart from acute myocardial infarction and heart failure, these diseases include valve conditions. Globally, around 300,000 heart valve replacement surgeries are performed annually, a number expected to rise to 850,000 by the year 2050 [1]. Surgical replacement using artificial heart valves is the gold standard despite several shortcomings. Tissue engineering possesses the potential to overcome these challenges by producing a heart valve closely mimicking the native tissue. However, no approach has been found that can mimic the mechanical properties of the native heart valve yet [2].

The mechanical properties of native heart valves are highly anisotropic [3]. This is mainly due to the three-layered histological composition of heart valve leaflets, comprising fibrosa, spongiosa, and ventricularis [4]. A load-oriented arrangement of collagen bundles accounts for the fibrosa’s high stress resistance. Collagen is the main load-bearing component of the heart valve tissue, and its anisotropic arrangement allows the valve to withstand high systolic pressure and a high number of load cycles (90,000 load cycles per day). In addition to strength, elasticity is essential to ensure the flexibility required for proper valve opening and closing behavior [5]. Radially aligned elastin bundles in the ventricularis confer this elasticity to the heart valve leaflets and play a critical role in achieving optimal closing behavior.

Early approaches to tissue-engineered heart valves (TEHVs) utilized isotropic scaffolds and relied on in vitro cultivation to induce anisotropic extracellular matrix (ECM) development. However, ECM growth is a slow process, and the strength developed after 4 weeks of cultivation falls short of native tissue. Consequently, newer approaches employ anisotropic scaffolds that can bear the required load while upholding the hemodynamic properties mandated by DIN EN ISO-5840.

In recent years, different manufacturing techniques for biomimetic scaffold designs were proposed to mirror the native mechanical behavior [6,7,8]. The technologies proposed were electrospinning, 3D printing, and melt electro-writing. All three production methods can achieve the load-dependent mechanical behavior and ad hoc mechanical strength typical of native valvular tissue [9,10]. However, most of the studies have predominantly focused on either the scaffold’s mechanical properties or hemodynamic characterization [11].

In our lab, multiple iterations of heart valve scaffolds have been proposed. A biohybrid aortic valve scaffold with biomimetic reinforcement was successfully cultivated and was able to withstand aortic flow and pressure conditions [12]. This scaffold had a three-leaflet design. This type of design requires a high degree of manual labor and open-heart surgery for implantation. In a prior study conducted in our lab, we proposed a tubular scaffold design suitable for transcatheter implantation [13,14]. We demonstrated that crimping and re-expansion had no significant effect on valve function. The tubular design reduces the number of production steps and does not require suturing or gluing individual leaflets, thereby reducing susceptibility to calcification [15]. This approach aligns with recent studies and represents the prevailing trend.

From the experience gained in our lab and the most recent approaches in the literature, we proposed that a biomimetically reinforced e-spun tubular scaffold will yield a heart valve scaffold that fulfills the hemodynamic requirements of DIN EN ISO 5840 while also possessing the necessary mechanical strength. To this end, we have developed a novel manufacturing process that combines solution electrospinning with reinforcement fibers. The produced fiber-reinforced scaffolds were tested for their mechanical and hemodynamic properties under physiological conditions and were compared against non-reinforced polycarbonate urethane (PCU) and native heart valve leaflets of porcine origin.

## 2. Materials and Methods

### 2.1. Scaffold Production

To investigate the production parameters, variations were introduced in the spinning solution concentration (8–12 wt%), potential difference (24–28 kV), and distance to the collector (5–15 cm). The polycarbonate urethane (PCU), (Carbothane^®^ PC3575A (Lubrizol Advanced Materials Inc., Cleveland, Ohio, USA), MW = 217 ± 2 kDa), was supplied in pellet form by Velox GmbH (Hamburg, Germany). The solution was prepared using a mixture of 75% chloroform and 25% methanol (both from VWR International GmbH (Darmstadt, Germany)) 24 h prior to electrospinning under continuous stirring. During the electrospinning process, a syringe pump delivered the solution at a rate of 2 mL/h. The process was carried out in a climate-controlled cabinet (23 °C and a humidity of 22%).

The scaffold production consisted of a three-step process: Initially, PCU was e-spun for 30 min (layer thickness 150 µm) onto a mandrel designed for this purpose. Subsequently, polyvinylidene fluoride (PVDF) monofilaments of the USP size 7-0 (diameter 0.5–0.69 mm) from G. Krahmer GmbH (Buchholtz, Germany) were arranged in the pattern depicted in Figure 1A. This arrangement involved securing six fibers to a single screw on the commissure side (F1 in Figure 1B) and draping them around the mandrel to create the developed pattern. Following this, two fibers each were fixed on opposing sides of a screw on the annulus side (F2 in Figure 1B). Following fiber placement, a second PCU e-spun layer (150 µm 30 min) was added to fix the fibers in place. A schematic section of the scaffold is depicted in Figure 1E. Figure 1B provides an image of the mandrel after winding, while Figure 1C exhibits the completed scaffold. Both materials were chosen based on their biocompatibility. Tschoeke et al. [15] tested PVDF as a standalone material. In addition, PVDF showed promising results when used for a vascular graft [16]. Uiterwijk et al. used PCU in an in vivo study with good results [17].

### 2.2. Scanning Electron Microscopy (SEM)

Scaffolds were sputter-coated (EM SC D500, Leica, Wetzlar, Germany) with a 12.5 nm gold-palladium layer and analyzed by scanning electron microscopy (ESEM XL 30 FEG, Philips, Hamburg, Germany) operating at an accelerating voltage of 10 kV. The fiber diameter and pore size were assessed using ImageJ freeware (National Institute of Health, Bethesda, MD, USA) by measuring the diameter of ten randomly selected fibers from three different SEM images. A representative SEM image is shown in Figure 1D. 

### 2.3. Biaxial Tensile Testing

The mechanical properties of the non-reinforced PCU, fiber-reinforced scaffold and native heart valve leaflets of porcine origin were evaluated by biaxial cyclic stress on a biaxial tensile tester (Zwick Roell GmbH & Co. KG, Ulm, Germany) fitted with custom-made clamps. For the biaxial tests, three fiber-reinforced scaffolds and three non-reinforced PCU samples were cut in a radial direction and were each divided into three leaflets, resulting in 25 × 25 mm samples (Figure 2B,D). Nine native heart valve leaflets, sourced from freshly slaughtered pigs (30–40 weeks old), were also tested. Due to the non-uniform cross-section of the fiber-reinforced scaffold, the force was normalized to the unstretched length of the sample. This results in the unit N/m, referred to as membrane tension, in line with previous work by Sacks et al. [18]. The results were calculated for each direction.

The test protocol was set to five cycles of physiological strain—10% (2.5 mm/min) in the circumferential and 30% (7.5 mm/min) in the radial direction [16]. The strain rates were adjusted to ensure that maximal deformation in both directions occurred simultaneously (Figure 2A). The testing speeds for native heart valve leaflets of porcine origin were customized based on each leaflet’s dimensions. Following five test cycles, a destructive test was performed on the same sample under equi-biaxial stretch at a strain rate of 50 mm/min to determine the non-directional mechanical strength. All reported values were calculated at the conclusion of each test section. The Young modulus was calculated within the linear range between 20 and 80 percent of the applied strain.

### 2.4. Burst Strength Testing

Burst strength tests were performed in a custom-made burst-strength setup equipped with a pressure sensor (Jumo Midas pressure transmitter; JUMO GmbH & Co. KG, Fulda, Germany) and a peristaltic pump (IPC Ismatec, IDEX Health & Science GmbH, Middleboro, MA, USA). Three samples were cut from each of the three fiber-reinforced scaffolds. The samples were subjected to increasing pressure on a 1 × 1 cm area by pumping buffered saline solution (PBS) at a constant rate of 5 mL/min. The burst strength was determined as the highest pressure measured until structural failure, as measured by a custom LabVIEW (National Instruments, Austin, TX, USA) program.

### 2.5. Bending Stiffness

Bending stiffness was assessed using the cantilever procedure for non-woven fabrics, according to DIN EN ISO 9073-7. Briefly, the fiber-reinforced scaffolds (n = 3) were cut into rectangular testing samples (length = 7.5 cm, width = 2 cm) and tested under wet conditions. The samples were shoved in a radial direction above the edge of a horizontal shelf until the sample tip was bent under an angle of 41.5° to the shelf. Subsequently, the overhang length (l_o_) was measured, and the bending stiffness (B) was calculated according to the formula:$$B=m\times {\left(l_{o}÷2\right)}^{3}$$
B = Bending stiffness; m = mass; l_o_ = overhang length

### 2.6. Hemodynamic Characterization

Three fiber-reinforced scaffolds were sutured into a silicon model of a human aortic root using the single-point attachment commissure technique, involving a continuous suture line at the annulus and a single stitch around the nearest fiber crossing for the commissure. The valves were tested in a custom-made flow-loop system, capable of applying physiological conditions according to ISO standard 5840 (cardiac output of 5.0 L/min; 100 mmHg mean ventricular pressure (120/80 mmHg) and 70 bpm frequency). The pressure was measured using transducers placed upstream and downstream from the valve (DPT 6000, pvd CODAN Critical Care GmbH, Forstinningen, Germany), while the flow was measured by a flowmeter (sonoTT, em-tec GmbH, Finningen, Germany) and recorded by a custom LabVIEW program (National Instruments, Austin, USA). The pressure difference between the atrium and the ventricle when the valve was open was defined as the mean gradient pressure. The regurgitation fraction and EOA were calculated according to ISO 5840–2. All values were calculated from ten cycles after achieving aortic conditions for at least 10 min.

### 2.7. Statistical Analysis 

The result data were expressed as mean ± standard deviation (SD). Comparisons between groups were performed using an ANOVA test followed by a post hoc test. The statistical significance was set at *p* < 0.05 unless stated otherwise. 

## 3. Results

### 3.1. Mandrel Design

The objective was to design a fiber-reinforced scaffold for tube-in-stent implantation on the basis of the previously developed and tested three-leaflet design. The tube had a length of 20 mm, and the length of the leaflet was 15 mm. Using the modeling results from Reese et al. [17,18] to develop a load-oriented reinforcement pattern, we designed a mandrel as a template for the fiber placement (Figure 1B). However, additional geometric constraints were necessary for a well-defined system. We established guidelines for implementing the reinforcement fiber configuration:Reinforcement fibers cross with an angle between 90° and 120°, following Stapelton et al.’s finding of optimal stress distribution within the scaffold [19];There is a concentric accumulation of the reinforcement fibers at the commissural points;Fiber crossings are incorporated within the leaflet region;A three-leaflet architecture is employed (Figure 1A);Only reinforce one leaflet.

We chose to prioritize fiber accumulation at the commissure points while adhering to the specified angle range. The fiber angle, location of fiber crossings, and accumulation points are influenced by the following parameters:The wrapping angle of the reinforcement fiber around the collector;The distance between fiber deflection points (annulus side);The distance between the scaffold area and fiber deflection points.

Moreover, the method of securing reinforcement fibers in place has an impact on how the pattern can be realized. Therefore, it has to be selected before the design of the collector. We considered the following options: (i) screws, (ii) glue and (iii) needles. Due to concerns over the solvent-related risks, we opted for screws as deflection points. Achieving concentric accumulation requires one deflection point on the commissure side and multiple deflection points on the annulus side to separate the reinforcement fibers. Given that our previous work used five fibers in each direction for leaflet reinforcement, there could be either three deflection points (two fibers per deflection point) or five deflection points (one per fiber) on the annulus side. Each additional deflection point increases the distance from the reinforcement fiber to the scaffold. With one deflection point per fiber, the difference in angle becomes too big. To maintain a consistent fiber angle, we selected three deflection points on the annulus side.

Calculating the ideal placement of deflection points involved a simplified (rectangular) leaflet shape. To achieve a 90° fiber crossing angle, the length of the circular arch had to match the distance between the fiber deflection points in the radial direction. With a design height of 20 mm and a leaflet width of 23.5 mm, deflection points should be placed 1.75 mm above and below the scaffold area. This, however, is not possible, given that the screws would damage the scaffold itself. Therefore, the wrapping angle had to be increased; otherwise, the angle would drop below 90°. The minimal and maximal fiber angles were computed using two fixed points:6.Commissures;7.Either the midpoint of the leaflet (maximal angle) or the neighboring leaflet’s edge (minimal angle).

With these two fixtures, we were able to calculate the possible fiber angles, whereas the minimal crossing angle is 70° and the maximal crossing angle is 116°. Placing the deflection points 120° from the leaflet on the annulus side, and 60° on the commissure side led to the best accumulation. A lower wrapping angle destroys the symmetry of the mandrel, and a higher wrapping angle would have only increased the distance between deflection points and the scaffold, which is an undesirable outcome. 

To sum up, we found that a fiber wrapping of 300° with an asymmetrical division—three deflection points on the annulus side and one on the commissure side, with the geometry displayed in Figure 1A—achieved the closest biomimetic pattern.

### 3.2. Electrospinning

We performed a systematic investigation to determine the optimal production parameters for the e-spun scaffold. Details of all tested parameter combinations, along with the resulting fiber diameter, pore size, and SEM images, are provided in the Appendix A. Pore sizes (ranging from 3.15 ± 0.06 to 5.31 ± 0.10 µm) and fiber diameters (ranging from 0.97 ± 0.35 to 1.76 ± 0.85 µm) exhibited similar orders of magnitude across all tested parameter combinations. The combination of a 26 kV potential difference, 10 wt% PCU and a 10 cm distance to the collector yielded the most stable spinning process. A representative SEM image is presented in Figure 1E.

### 3.3. Mechanical Testing

The results of the cyclic tensile tests conducted on fiber-reinforced scaffolds, non-reinforced PCU, and native heart valve leaflets of porcine origin are depicted in Figure 3A–F. In the radial direction, the membrane tension after the first cycle of the fiber-reinforced scaffold significantly exceeded that of the subsequent cycles (*p* < 0.01). However, no significant differences in membrane tension were observed in cycles 2–5. In the circumferential direction, the membrane tension in cycle 1 was significantly higher (*p* < 0.01) than in cycles 3–5. Furthermore, the tension measured in cycle 2 was significantly higher than in the following cycles. The mechanical behavior of non-reinforced PCU exhibited similarity to that of fiber-reinforced scaffolds in the radial direction, with a significant decrease observed from cycle 1 to the subsequent cycles (*p* < 0.01) but no differences in cycles 2–5. Similar results were noted in the circumferential direction for non-reinforced PCU. Throughout the cyclic biaxial stretching, no significant differences in membrane tension were detected between the fiber-reinforced scaffolds and non-reinforced PCU.

When comparing fiber-reinforced scaffolds with native heart valve leaflets of porcine origin, the radial-direction membrane tension was significantly higher over all five cycles (Figure 3D,F). In the circumferential direction, the fiber-reinforced scaffold exhibited a significantly higher membrane tension after cycle 1 (*p* < 0.01) than native heart valve leaflets of porcine origin ((139.59 ± 4.68 N/m vs. 49.74 ± 6.93 N/m). However, the native heart valve leaflets of porcine origin displayed greater elasticity. Following five cycles, the membrane tension of the native heart valve leaflets of porcine origin was significantly higher (41.02 ± 7.68 N/m) compared to the fiber-reinforced scaffold (23.41 ± 1.22 N/m) (Figure 3A,C).

The equi-biaxial test resulted in the rupture of the fiber-reinforced scaffold in the circumferential direction at a strain of 53.1 ± 9.24%, a membrane tension of 461.64 ± 58.87 N/m and a Young modulus of 8.71 ± 1 N/m. In the radial direction, the membrane tension was 328.25 ± 20.36 N/m, with a Young modulus of 6.19 ± 0.38 N/m. While the reinforcement fibers did not fail, the PCU was detached from the fibers. Compared to the native heart valve of porcine origin, the fiber-reinforced scaffold exhibited significantly higher mechanical strength, with a membrane tension of 461.64 ± 58.87 to 165.52 ± 17.23 N/m in the circumferential direction and 328.25 ± 20.36 to 146.53 ± 12.68 N/m in the radial direction. Additionally, the Young modulus was also higher for the fiber-reinforced scaffold compared to native heart valve leaflets of porcine origin (8.71 ± 1.1 to 4.42 ± 0.98 N/m in the circumferential direction and 6.19 ± 0.38 to 2.62 ± 0.58 N/m in the radial direction (Figure 4)). Nonetheless, native heart valve leaflets of porcine origin are not quadratic, resulting in a significantly lower strain of 37.6 ± 6.52% in the circumferential direction compared to the fiber-reinforced scaffold.

The fiber-reinforced scaffold possesses a burst strength of 1402 ± 324 mmHg. The average bending stiffness for the tested samples was determined through the Cantilever procedure as 0.52 ± 0.5 mNcm^2^.

### 3.4. Valve Fashioning and Hemodynamic Characterization

Commissures were placed at the first reinforcement fiber crossing point, approximately 5 mm below the upper edge, and spaced at 120° ± 10° intervals along the circumference. A commissure positioned at the scaffold’s upper edge without the inclusion of a reinforcement fiber instantly failed, as did all attempts to test a non-reinforced PCU valve.

A representative opening and closing cycle under aortic conditions is shown in Figure 5A–E. The average opening time was 0.49 ± 0.03 s per cycle. The mean transvalvular pressure gradient recorded was 3.02 ± 0.26 mmHg, and the effective orifice area (EOA) was determined to be 2.12 ± 0.22 cm^2^, exceeding the minimum required EOA of 1.45 cm^2^ for a 25 mm heart valve The transvalvular regurgitant fraction was 11.51 ± 1.25%, which is below the required 15%. The TEHVs were not tested to the point of failure. The tests were abandoned after 4 h at 70 BPM.

## 4. Discussion

Although heart valve tissue engineering has been a subject of research for decades, the creation of a well-functioning valve for systemic circulation (mitral and aortic positions) with adequate long-term mechanical fatigue properties remains a challenge [20]. This study introduces a novel method to manufacture a tubular biomimetic textile-reinforced tissue-engineered heart valve (TEHV) scaffold for the aortic position. The fiber-reinforced scaffold comprises PVDF reinforcement fibers arranged in a biomimetic pattern and held in place by e-spun PCU (Figure 1A–C). Through the development of a specialized mandrel, combining an electrospinning collector and the placement pattern for the reinforcement fibers, a standardized manufacturing process was established to create a load-oriented anisotropic aortic heart valve scaffold.

Prior work has mainly centered on the orientation of e-spun fibers or only used reinforcement fibers in the leaflets rather than throughout the entire scaffold [10,21]. Electrospinning is easily scalable and can be standardized. Scalability and reproducibility are two points of emphasis in the clinical approval process. Although studies have demonstrated satisfactory outcomes for pulmonary and aortic conditions [11,22,23], the number of investigations assessing the hemodynamics of TEHVs in the aortic position remains limited. Most of these studies have primarily concentrated on the mechanical and microscopic properties [10,24].

In the literature, the physiological membrane tension is reported as 60 N/m [16]. However, our lab’s tests yielded a membrane tension of 45 N/m for native heart valves of porcine origin. After five cycles of physiological strain (30% in the radial and 10% in the circumferential directions), the produced fiber-reinforced scaffold exhibited a membrane tension of 69.45 ± 9.02 N/m in the radial direction and 23.41 ± 1.22 N/m in the circumferential direction. The non-reinforced PCU showed membrane tensions of 67.25 ± 7.1 N/m in the radial and 33.45 ± 7.9 N/m in the circumferential direction after five cycles of physiological strain. Both specimens surpassed the physiological membrane tension reported in the literature (60 N/m) in a radial direction after five cycles but not in a circumferential direction (60 N/m) [16]. Ravishankar et al. developed a scaffold by electrospinning a polycaprolactone gelatin blend, subsequently coated with methacrylated gelatin, achieving an ad hoc strength of 90 N/m in both directions during physiological stretches [10]. The fiber-reinforced scaffold produced in this study processes an ad hoc strength 1.5 times that of the scaffold produced by Ravishankar et al. Considering that a heart valve undergoes 90,000 load cycles per day, we propose that a cyclic test delivers more relevant results. Nevertheless, only a fatigue test can reveal the long-term strength of the produced scaffolds. While the measured stresses in the circumferential direction after five cycles fall below the physiological values reported in the literature [16], the failure stress is five times that of the physiological stresses (60 N/m in both directions). The cutting of the scaffold into three leaflets also damages and weakens the reinforcement structure in the circumferential direction.

Moreover, it can be inferred that the reinforcement fibers do not bear much stress during physiological strain following their initial stretching in cycle one. The Young modulus of the fiber-reinforced scaffold is initially higher in the circumferential direction and then equalizes over the subsequent four cycles to the values measured for the non-reinforced PCU. After five cycles, the Young modulus becomes equal in both the radial and circumferential directions for both specimens, indicating no significant difference in mechanical strength (membrane tension and Young modulus) between the fiber-reinforced scaffold and non-reinforced PCU. However, when tested to the point of failure, there is a significant difference between the fiber-reinforced scaffold and non-reinforced PCU, suggesting that the reinforcement fibers protect the e-spun PCU from overloading. This is corroborated by the failure mechanism: while the fiber-reinforced scaffold fails in the circumferential direction, the non-reinforced PCU tears in multiple places in both directions (Figure 2E,F).

As a standalone material, the PCU already facilitates cell integration, and extracellular matrix components can be deposited in the PCU scaffold [25,26]. The electrospun PCU emulates the native fibrillar environment of the native heart valve. The e-spun fiber diameter (1.47 ± 0.55 µm) is within the range of native ECM components (0.2–2 µm) [27,28]. However, reports suggest that the ideal fiber diameter for cell integration in cardiovascular tissue engineering is 5–10 µm [29,30,31]. Cell integration can also be enhanced through a hydrogel coating, such as Matrigel or poly(ethylene glycol) methacrylate [24,32]. In a previous study conducted in our lab, we produced a scaffold with a three-leaflet design. By doing this, we showed that a combination of load-oriented textile reinforcement and cell-laden fibrin produced a considerable amount of collagen while remaining pliable enough to function as an aortic valve [12,14].

In comparison to previously published scaffolds, the presented valve demonstrates further improvement in all pertinent mechanical properties. Due to alterations in the reinforcement pattern and material composition, the bending stiffness of the scaffold decreased from 5.9 ± 1.2 mNcm^2^ to 0.52 ± 0.5 mNcm^2^ in the radial direction. Reduced bending stiffness enhances the flexibility of the valve leaflets, which is essential for proper functionality. A notable standard deviation was observed in the sample measurements, suggesting substantial variations in the mechanical properties among the e-spun fiber-reinforced scaffolds. However, even the stiffest samples (1.29 mNcm^2^) exhibited decreased bending stiffness compared to the scaffold from Moreira et al. (5.9 ± 1.2 mNcm^2^) [12].

While the physiological burst strength is 2473 ± 230 mmHg, we managed to improve the values previously measured in our lab from 1086 mmHg to 1402 ± 324 mmHg, which is about tenfold the systolic pressure [13]. Hence, the mechanical properties of the produced scaffold were sufficient to function as an aortic valve without prior cell culture.

The hemodynamic characteristics of the fiber-reinforced scaffold align with the requirements defined by ISO 5840-2 regarding the efficient opening area (EOA) and transvalvular regurgitant fraction. For a 25 mm valve, the EOA required by ISO 5840 requires an EOA of 1.45 cm^2^ and a maximum transvalvular regurgitant fraction of 15%. The manufactured scaffold exhibits an EOA of 2.12 ± 0.22 cm^2^ and a transvalvular regurgitant fraction of 11.51 ± 1.25%. Furthermore, the produced heart valve demonstrates a low transvalvular pressure gradient of 3.02 ± 0.26 mmHg, which indicates very good pliability of the fiber-reinforced scaffold. This pressure gradient is lower in comparison to a stentless pericardial valve implanted in sheep, as demonstrated in a study by Goetz et al. (with an initial gradient of 3.7 ± 2.2 mmHg) [33]. Both the EOA and pressure gradient compare favorably with the clinical studies by Filip et al. and Johnson et al. of a tube-in-stent aortic heart valve derived from bovine or porcine pericardium. These studies reported a maximum EOA of 1.8 cm^2^ postoperatively, and the mean pressure gradient during the follow-up period did not drop below 7 mmHg [34,35].

The clinical significance of tissue-engineered heart valves (TEHVs) could be notably enhanced by the development of in situ transcatheter tissue-engineered aortic valve substitutes [15]. This approach offers a minimally invasive delivery method with reduced risks compared to the open-heart surgery required for mechanical valves.

The scaffold design detailed in this paper represents a promising advancement toward an in situ tissue engineering strategy, as it demonstrates the ability to endure acute aortic conditions without requiring in vitro cultivation [36,37]. To thoroughly assess the in situ potential of the developed scaffold, the logical next step would involve conducting an in vivo study.

## 5. Conclusions

In this study, we succeeded in designing and producing a fiber-reinforced scaffold that fulfills the hemodynamic criteria according to ISO 5840 for an aortic heart valve in acute tests and has an excellent transvalvular pressure gradient. In addition, the mechanical properties compare well to native heart valves of porcine origin. Therefore, this scaffold is a good candidate for an in situ implantation.

## Figures and Tables

**Figure 1 bioengineering-10-01064-f001:**
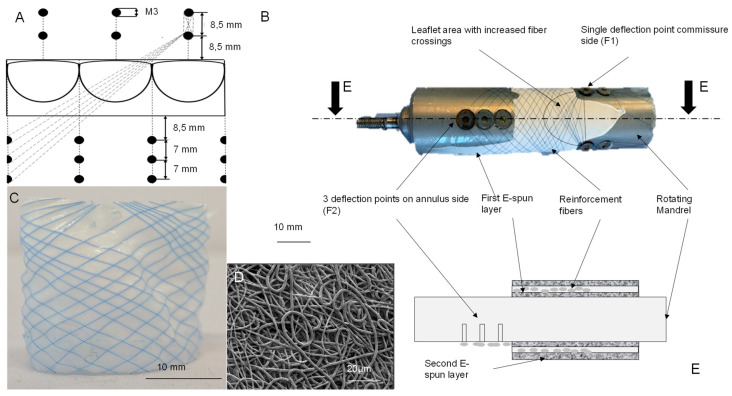
Design of the fiber-reinforced scaffold and mandrel of the commissure side is towards the top of the images: (**A**) Geometrical alignment of anchor points on the mandrel; (**B**) mandrel with one layer of polycarbonate polyurethane (PCU) and polyvinylidene fluoride (PVDF)-reinforcement fibers; (**C**) finished fiber-reinforced scaffold; (**D**) scanning electron microscopy (SEM) image e-spun PCU with fabrication settings (10 wt%, 26 kV and 10 cm distance between needle and mandrel); (**E**) section of the fiber-reinforced scaffold at the end of the production process.

**Figure 2 bioengineering-10-01064-f002:**
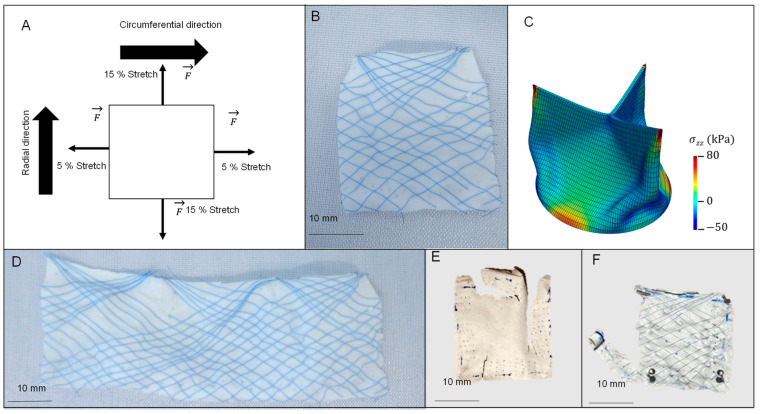
Biaxial test of the polycarbonate polyurethane (PCU) and fiber-reinforced scaffold: (**A**) schematic test procedure of the biaxial test; (**B**) leaflet cut from the tubular fiber-reinforced scaffold to be tested; (**C**) simulated mechanical stress in closed position; (**D**) fiber-reinforced scaffold cut open in radial direction, all three leaflets are visible; (**E**) failed non-reinforced PCU sample after destructive test; (**F**) failed fiber-reinforced scaffold leaflet after destructive test. The direction shown in A applies to all other images as well.

**Figure 3 bioengineering-10-01064-f003:**
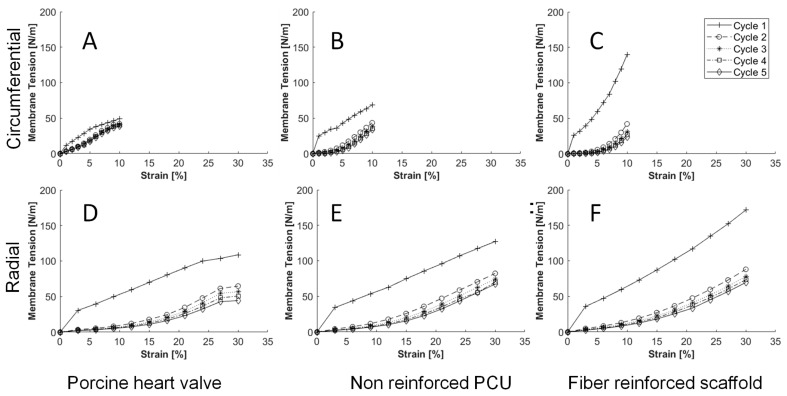
(**A**) Mechanical test results of the native leaflet in circumferential direction; (**B**) mechanical test results of the non-reinforced polycarbonate polyurethane (PCU) in circumferential direction; (**C**) mechanical test results of the scaffold in circumferential direction; (**D**) mechanical test results of the native leaflet in radial direction; (**E**) mechanical test results of the non-reinforced PCU in radial direction; (**F**) mechanical test results of the scaffold in radial direction; *x*-axis strain [%] y-axis membrane tension [N/m].

**Figure 4 bioengineering-10-01064-f004:**
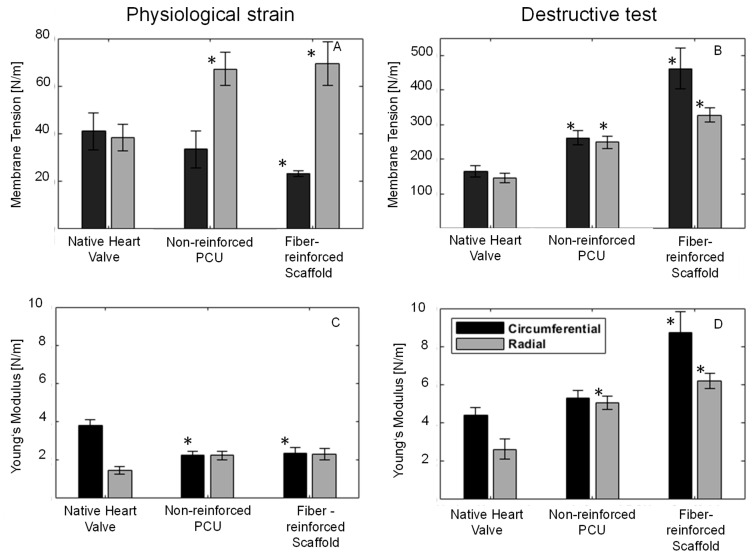
(**A**) Membrane Tension of the native heart valve, non-reinforced polycarbonate polyurethane (PCU) and fiber-reinforced scaffold at physiological strains (10% in circumferential direction and 30% in radial direction) at the end of the cyclic tests; (**B**) Membrane Tension of the native heart valve, non-reinforced PCU and fiber-reinforced scaffold at the point of destruction. (**C**) Young modulus of the native heart valve, non-reinforced polycarbonate polyurethane (PCU) and fiber-reinforced scaffold at physiological strains (10% in circumferential direction and 30% in radial direction) at the end of the cyclic tests; (**D**) Young modulus of the native heart valve, non-reinforced PCU and fiber-reinforced scaffold at the point of destruction. * marks a significant difference compared to the native heart valve (9 samples each were tested).

**Figure 5 bioengineering-10-01064-f005:**
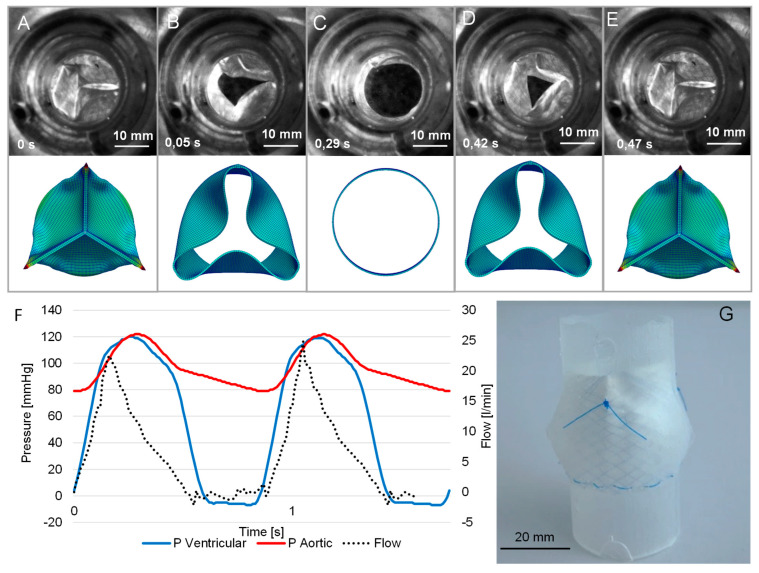
Hemodynamic characterization of a fashioned valve: (**A**–**E**) typical opening and closing cycle of the heart valve under aortic conditions with simulated mechanical stress; (**F**) aortic and ventricular pressure over two cycles at 70 BPM; (**G**) fashioned valve in silicon aorta.

## Data Availability

All research date will be made available upon request.

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
