# Peer review of "Bio-Inspired Fiber Reinforcement for Aortic Valves: Scaffold Production Process and Characterization"

_bioengineering, 2023, doi:10.3390/bioengineering10091064_

Round 1

Reviewer 1 Report

This manuscript developed a new manufacturing process to fabricate a fiber-reinforced biomimetic scaffold for aortic heart valve tissue engineering. The reinforcement structure of polyvinylidene difluoride monofilament fibers are biomimetically arranged and embedded into electro-spun polycarbonate urethane. The target scaffold was characterized by bi-axial tensile strength, bending stiffness, burst pressure and hemodynamically in a mock circulation system.

The results showed these values were above the levels of tested native heart valve tissue, which is promising for valve replacement. However, there’s still improvement space for the current manuscript.

1. The durability of the heart valve prostheses should be tested, which could be regarded as one of the most important assessment criteria according to the ISO standard 5840:2015. The cycle numbers without failure should be provided, as well as the equivalent working time inside the human body.

2. The statistical method is missing in the Materials and Methods parts. The replicate numbers should be provided for Figure 4, and statistical analysis should be performed to assess significant differences between the control and test groups.

Acceptable

Author Response

Dear Reviewer,

Thank you for evaluating our manuscript and considering it for publication. We carefully read your remarks and addressed them in the following way:

Point 1: The durability of the heart valve prostheses should be tested, which could be regarded as one of the most important assessment criteria according to the ISO standard 5840:2015. The cycle numbers without failure should be provided, as well as the equivalent working time inside the human body.

Response 1: Thank you for pointing out the lack of fatigue tests in our manuscript. Evaluating the long term durability is indeed a very important aspect of ISO 5840. Fatigue tests are ongoing and we plan to publish those results separately together with results from a planned in-vivo study. Nevertheless, in this study the focus was on the development of the mandrel and production process as a proof of concept.

Point 2: The statistical method is missing in the Materials and Methods parts. The replicate numbers should be provided for Figure 4, and statistical analysis should be performed to assess significant differences between the control and test groups.

Response 2: A paragraph statistical analysis was added to the methods section and the analysis of the young’s moduli was indicated in Fig. 4

“Results data were expressed as mean ± standard deviation (SD). Comparisons between groups were performed by ANOVA test followed by a post hoc test. Statistical significance was set at p < 0,05 unless stated otherwise.”

Please see below the remarks issued by the second reviewer and how they were addressed:

Points 1 and 2: We request you to kindly go through the manuscript thoroughly. There are lot of places where the sentences don’t sound right. For example, line 206, line 277. Also, there are lot of sentences, where double space is observed between two words. For example, line 193. It is requested the authors to go through the manuscript thoroughly.

Response 1 and 2: The Manuscript was carefully revised regarding language and double spacing.

Point 3: Have the authors conducted any biocompatibility or biodegradability test of the aortic scaffold manufactured? It would be good to add this to the study to increase the value of the paper for commercial visibility.

Response 3: The materials were chosen based on studies indicating good biocompatibility of PVDF and PCU. In our lab, we successfully produced vascular grafts fabricated from PVDF [1, 2]. Uiterwijk et al. published a promising in-vivo study with an e-spun PCU scaffold [3]. We added a paragraph to the materials and methods section explaining our choice of materials.

„Both materials were chosen based on their biocompatibility, Tschoeke et al. [2] tested PVDF as a standalone material. In addition PVDF showed promising results when used for a vascular graft [1]. Uiterwijk et al. used PCU in an in-vivo study with good results [3]“

Point 4: The supplementary information contains only the image. It does not contain figure caption.

Response 4: We re-arranged the supplementary information by dividing the image into three and added a caption to each.

We appreciate your time and effort in reviewing our manuscript and hope this addresses your comments appropriately.

Kind Regards

Christian Boehm

1              Wolf, F., et al., MR and PET-CT monitoring of tissue-engineered vascular grafts in the ovine carotid artery. Biomaterials, 2019. 216: p. 119228.

2              Tschoeke, B., et al., Development of a composite degradable/nondegradable tissue-engineered vascular graft. Artif Organs, 2008. 32(10): p. 800-9.

3              Uiterwijk, M., et al., In Situ Remodeling Overrules Bioinspired Scaffold Architecture of Supramolecular Elastomeric Tissue-Engineered Heart Valves. JACC Basic Transl Sci, 2020. 5(12): p. 1187-1206.

Reviewer 2 Report

Review comments

The research paper entitled, ‘Bioinspired fiber-reinforcement for aortic valves; scaffold production process and characterisation’ demonstrates a novel method to manufacture bioinspired fiber-reinforcement for an aortic valve scaffold.

The authors demonstrate a hybrid method of manufacturing with tremendous commercial value.

The reviewer recommends the paper for publication with minor revision.

1.      We request you to kindly go through the manuscript thoroughly. There are lot of places where the sentences don’t sound right. For example, line 206, line 277.

2.      Also, there are lot of sentences, where double space is observed between two words. For example, line 193. It is requested the authors to go through the manuscript thoroughly.

3.      Have the authors conducted any biocompatibility or biodegradability test of the aortic scaffold manufactured? It would be good to add this to the study to increase the value of the paper for commercial visibility.

4.      The supplementary information contains only the image. It does not contain figure caption.

Author Response

Dear Reviewer,

Thank you for evaluating our manuscript and considering it for publication. We carefully read your remarks and addressed them in the following way:

Points 1 and 2: We request you to kindly go through the manuscript thoroughly. There are lot of places where the sentences don’t sound right. For example, line 206, line 277. Also, there are lot of sentences, where double space is observed between two words. For example, line 193. It is requested the authors to go through the manuscript thoroughly.

Response 1 and 2: The Manuscript was carefully revised regarding language and double spacing.

Point 3: Have the authors conducted any biocompatibility or biodegradability test of the aortic scaffold manufactured? It would be good to add this to the study to increase the value of the paper for commercial visibility.

Response 3: The materials were chosen based on studies indicating good biocompatibility of PVDF and PCU. In our lab, we successfully produced vascular grafts fabricated from PVDF [1, 2]. Uiterwijk et al. published a promising in-vivo study with an e-spun PCU scaffold [3]. We added a paragraph to the materials and methods section explaining our choice of materials:

„Both materials were chosen based on their biocompatibility, Tschoeke et al. [2] tested PVDF as a standalone material. In addition PVDF showed promising results when used for a vascular graft [1]. Uiterwijk et al. used PCU in an in-vivo study with good results [3]“

Point 4: The supplementary information contains only the image. It does not contain figure caption.

Response 4: We re-arranged the supplementary information by dividing the image into three and added a caption to each.

Please see below the remarks issued by the second reviewer and how they were addressed:

Point 1: The durability of the heart valve prostheses should be tested, which could be regarded as one of the most important assessment criteria according to the ISO standard 5840:2015. The cycle numbers without failure should be provided, as well as the equivalent working time inside the human body.

Response 1: Thank you for pointing out the lack of fatigue tests in our manuscript. Evaluating the long term durability is indeed a very important aspect of ISO 5840. Fatigue tests are ongoing and we plan to publish those results separately together with results from a planned in-vivo study. Nevertheless, in this study the focus was on the development of the mandrel and production process as a proof of concept.

Point 2: The statistical method is missing in the Materials and Methods parts. The replicate numbers should be provided for Figure 4, and statistical analysis should be performed to assess significant differences between the control and test groups.

Response 2: A paragraph statistical analysis was added to the methods section and the analysis of the young’s moduli was indicated in Fig. 4

“Results data were expressed as mean ± standard deviation (SD). Comparisons between groups were performed by ANOVA test followed by a post hoc test. Statistical significance was set at p < 0,05 unless stated otherwise.”

We appreciate your time and effort in reviewing our manuscript and hope this addresses your comments appropriately.

Kind Regards

Christian Boehm

1              Wolf, F., et al., MR and PET-CT monitoring of tissue-engineered vascular grafts in the ovine carotid artery. Biomaterials, 2019. 216: p. 119228.

2              Tschoeke, B., et al., Development of a composite degradable/nondegradable tissue-engineered vascular graft. Artif Organs, 2008. 32(10): p. 800-9.

3              Uiterwijk, M., et al., In Situ Remodeling Overrules Bioinspired Scaffold Architecture of Supramolecular Elastomeric Tissue-Engineered Heart Valves. JACC Basic Transl Sci, 2020. 5(12): p. 1187-1206.

Round 2

Reviewer 1 Report

All of the comments were successfully addressed, and the manuscript was further improved.

Minor editing is required.